# The Genetic Diversity and Antimicrobial Resistance of Pyogenic Pathogens Isolated from Porcine Lymph Nodes

**DOI:** 10.3390/antibiotics12061026

**Published:** 2023-06-07

**Authors:** Aleksandra Kaczmarkowska, Ewelina Kwiecień, Anna Didkowska, Ilona Stefańska, Magdalena Rzewuska, Krzysztof Anusz

**Affiliations:** 1Department of Food Hygiene and Public Health Protection, Institute of Veterinary Medicine, Warsaw University of Life Sciences (SGGW), 02-787 Warsaw, Poland; anna_didkowska@sggw.edu.pl (A.D.); krzysztof_anusz@sggw.edu.pl (K.A.); 2Department of Preclinical Sciences, Institute of Veterinary Medicine, Warsaw University of Life Sciences (SGGW), 02-787 Warsaw, Poland; ewelina_kwiecien1@sggw.edu.pl (E.K.); ilona_stefanska@sggw.edu.pl (I.S.); magdalena_rzewuska@sggw.edu.pl (M.R.)

**Keywords:** antimicrobial resistance, genetic diversity, lymph nodes, pigs, pork, *R. equi*, *S. aureus*, *S. dysgalactiae*, *S. equi*, zoonoses

## Abstract

According to the Food and Agriculture Organization of the United Nations, pork remains the most consumed meat in the world. Consequently, it is very important to ensure that it is of the highest microbiological quality. Many of the pathogens that cause lymph node lesions in pigs are zoonotic agents, and the most commonly isolated bacteria are *Mycobacterium* spp., *Streptococcus* spp., *Staphylococcus aureus* and *Rhodococcus equi* (synonymous with *Prescottella equi*). The prevention and treatment of zoonotic infections caused by these bacteria are mainly based on antimicrobials. However, an overuse of antimicrobials contributes to the emergence and high prevalence of antimicrobial-resistant strains, which are becoming a serious challenge in many countries. The aim of this study was to evaluate the genetic diversity and antimicrobial resistance of the *Streptococcus* spp. (*n* = 48), *S. aureus* (*n* = 5) and *R. equi* (*n* = 17) strains isolated from swine lymph nodes with and without lesions. All isolates of *S. dysgalactiae*, *S. aureus* and *R. equi* were subjected to PFGE analysis, which showed the genetic relatedness of the tested bacteria in the studied pig populations. Additionally, selected tetracycline and macrolide resistance genes in the streptococcal strains were also studied. The results obtained in the present study provide valuable data on the prevalence, diversity, and antimicrobial resistance of the studied bacteria. Numerous isolated bacterial *Streptococcus* spp. strains presented resistance to doxycycline, and almost half of them carried tetracycline resistance genes. In addition, *R. equi* and *S. aureus* bacteria presented a high level of resistance to beta-lactam antibiotics and to cefotaxime, respectively.

## 1. Introduction

The assessment of the microbiological quality of food is an important step in providing safe food products. Due to the fact that pork is one of the most popular meats in the world, it is especially important to ensure that it is of the highest quality. During inspection at slaughterhouses, lesions of different organs may be observed, including suppurative lesions in lymph nodes. In the case of swine, lymph node lesions may be caused by various pathogens that frequently have zoonotic potential. The bacteria most often isolated from such lesions are the *Mycobacterium tuberculosis* complex (MTC), *Mycobacterium avium* (*M. avium*), *Rhodococcus equi* (synonymous with *Prescottella equi*) (*R. equi*), *Trueperella pyogenes* (*T. pyogenes*), *Staphylococcus aureus* (*S. aureus*) and various *Streptococcus* species [1,2,3,4].

According to the available data, the Gram-positive bacteria of the genus *Streptococcus* seem to be pathogens that are relatively often associated with suppurative lesions in humans and animals, including pigs [5]. *Streptococcus dysgalactiae* subsp. *equisimilis*, which is a pyogenic pathogen from Lancefield groups A, C, G and L, is increasingly being recovered from severe invasive infections in humans worldwide [6,7]. This bacterium can cause a broad spectrum of infections in humans, including skin abscesses, cellulitis, pharyngitis, arthritis, bacteremia, endocarditis and toxic shock syndrome [6,7]. *S. dysgalactiae* is part of the normal microbiota of pigs and is also considered an important pathogen, especially in piglets [7,8]. In contrast, another beta-hemolytic species, *Streptococcus equi* subsp. *zooepidemicus*, was identified as an emerging pig pathogen in 2019 and is associated with sudden deaths, increased miscarriages and septicemia [9]. In humans, these bacteria cause invasive infections in immunocompromised hosts [10,11,12].

Purulent infections in livestock, including pigs, may also be caused by *R. equi*. This aerobic, Gram-positive, opportunistic, and facultative intracellular pathogen was first isolated in 1923 from foals with pneumonia. It is widespread in the environment but is mainly isolated from suppurative lesions of various livestock species, including pigs. Moreover, *R. equi* is mentioned as one of the most important pathogens associated with lymph node infections in swine [13]. It should be noted that *R. equi* infections have also been increasingly diagnosed in immunocompromised persons in recent years, including patients who have received organ transplants [14,15].

Another pathogen that is sometimes involved in various multidrug-resistant infections worldwide is *S. aureus*. This bacterium is a Gram-positive coccus and is an opportunistic pathogen found in both humans and animals. It causes a range of diseases, including skin and soft tissue infections, infective endocarditis and toxic shock syndrome. These infections are often difficult to treat due to the multidrug resistance of *S. aureus* strains and are associated with high morbidity, mortality and a significant economic impact [16].

Swine lymph node lesions are a significant cause of the condemnation of pig meat during slaughterhouse inspections and thus lead to significant economic losses [1,13,17]. Although these lesions in pigs are often limited to the lymph nodes of the head, sometimes the lymph nodes and organs of the thorax or abdominal cavity are also affected [1,18,19]. Despite continuous efforts to increase food safety by improving the quality of processes, the confirmed presence of many types of pathogens, e.g., *S. aureus* or *S. dysgalactiae*, in retail pork still poses a significant risk to consumers [20,21,22]. The molecular characterization of bacterial strains isolated from animals in slaughterhouses may be one of the most important steps to achieving effective infection control. Pulsed-field gel electrophoresis (PFGE) is one of the best typing methods and has successfully been used to study the diversity of isolates of various species recovered from different clinical samples [23,24].

Moreover, the widespread use of antimicrobials has contributed to the evolutionary selection of bacterial strains that are resistant to various antimicrobial compounds. Antimicrobial-resistant bacteria are now frequently isolated from most species of farmed animals; therefore, this resistance is increasingly threatening the efficacy of drugs, including those historically used in humans [25].

The purpose of our study was to estimate the genetic relatedness of the pyogenic zoonotic pathogen strains *Streptococcus* spp., *R. equi* and *S. aureus*, which were isolated from swine lymph nodes. Moreover, the minimum inhibitory concentration (MIC) values of selected antimicrobial agents for the studied bacteria, as well as the tetracycline and macrolide resistance determinants in streptococci, were determined.

## 2. Results

### 2.1. Bacteriological Examination

The studied pathogenic isolates belonged to *Streptococcus* spp., *S. aureus* and *R. equi* (Table 1, Appendix A). Streptococci were identified in a significant number (48 out of 199; 24.1%) of the tested samples, including 31 lymph nodes and organs with lesions (31/48; 64.58%) and 17 lymph nodes without lesions (17/48; 35.4%). Using the rapid latex test, 14 streptococcal strains (14/48; 29.2%) were classified into Lancefield group A, and 34 (34/48; 70.8%) were classified into group C. The PCR results showed that 45 *S. dysgalactiae* strains (45/48; 93.8%) and three isolates (3/48; 6.2%) were identified as *S. equi* subsp. *zooepidemicus*. *R. equi* was isolated from 17 samples (17/199; 8.5%), including eight tissues with lesions (8/17; 47.1%) and nine lymph nodes without lesions (9/17; 52.9%). Based on the latex agglutination test and the PCR results, five isolates (5/199; 2.5%) were identified as *S. aureus*. One of them (1/5; 20%) was isolated from lymph nodes without lesions, and four (4/5; 80%) were isolated from tissue with lesions. Moreover, six coagulase-positive *Staphylococcus* spp. other than *S. aureus* were isolated: four from tissue with lesions and two from tissue without lesions. Additionally, *T. pyogenes* was isolated once from a mandibular lymph node with lesions. During the bacteriological examination, other bacteria were also isolated; however, due to their small number, they were not included in the analysis. These bacteria were coagulase-negative *Staphylococcus* spp., *Enterococcus* spp. and Gram-negative rods isolated from both lesions and tissues without lesions. Two strains of streptococci were isolated from individuals from whom more than two samples were taken, while three strains of streptococci and one strain of *R. equi* were isolated from an individual with generalized lesions.

### 2.2. Phenotypes and Genotypes of Antimicrobial Resistance

Antimicrobial susceptibility testing was carried out for 45 *S. dysgalactiae* strains, 3 strains of *S. equi* subsp. *zooepidemicus*, 17 *R. equi* strains, and 5 strains of *S. aureus*. The MIC value distributions (MIC_50_ and MIC_90_) of eight tested antimicrobials for the studied isolates are shown in Table 2. Additionally, the genotypes of resistance to tetracyclines and macrolides were determined for the *S. dysgalactiae* strains. The distribution of the minimum inhibitory concentration (MIC) of the eight antimicrobial agents for the tested isolates of *Streptococcus* spp., *R. equi* and *S.s aureus* from pigs is provided in Appendix A.

#### 2.2.1. Study for *S. aureus*

Among the tested *S. aureus* strains, all were susceptible to PEN, AMC, CIP, ERY, DOX, GEN and SXT. According to the used criterion, two strains (2/5; 40%) showed resistance to CTX, with MIC_50_ and MIC_90_ values of 12 µg/mL and >32 µg/mL, respectively. The remaining three strains (3/5; 60%) were classified as intermediate to CTX, with MIC_50_ and MIC_90_ values of 4 µg/mL and >32 µg/mL, respectively.

#### 2.2.2. Study for *Streptococcus* spp.

In the case of the *S. equi* subsp. *zooepidemicus* strains, their susceptibility to almost all the tested antimicrobial agents was found, including PEN, AMC, CTX, CIP, GEN, ERY and DOX. Two of the three tested strains (66.7%) were also susceptible to SXT, but one of them (1/3; 33.3%) was categorized as intermediate. All *S. dysgalactiae* strains (*n* = 45) were susceptible to the β-lactam antibiotics tested in this study, including PEN, AMC and CTX. Moreover, all strains were also susceptible to CIP and GEN. A similarly high susceptibility was observed for SXT: the vast majority of strains were susceptible (40, 88.9%), and a few were resistant (3, 6.7%) and intermediate (2, 4.4%). Four strains were resistant to ERY, all of them with an MIC value >256 µg/mL. More strains, 24 (50%) in total, were resistant to DOX, with an MIC ranging from 12 µg/mL to 64 µg/mL. Five of forty-five (11.1%) *S. dysgalactiae* strains (S1′, S27, S44, S61 and S115—two of which were isolated from the same herd) were concurrently resistant to 2 antimicrobials tested in this study (resistance phenotype: ERY/DOX). Generally, among the strains isolated from pigs from the same farm, differences in antimicrobial resistance profiles were noticeable.

Twenty-one of forty-five *S. dysgalactiae* strains (46.7%) were positive during the PCR with universal primers for the detection of tetracycline ribosome protection protein genes. Eighteen strains (40%) harbored the *tetM* gene, and three (6.8%) harbored *tetO*. Two strains, isolated from pigs from the same herd, carried both tetracycline resistance genes, *tetM* and *tetO*. In the case of one strain, the product of the reaction with universal primers was detected but not with the tested primers for specific genes. Four erythromycin-resistant strains (8.9%) were positive for the *ermB* gene. All *S. dysgalactiae* isolates tested in this study were negative for *tetK*/L and *ermA* genes.

#### 2.2.3. Study for *R. equi*

All 17 strains of *R. equi* (100%) showed resistance to PEN. However, 12 strains (70.6%) showed resistance to CTX, while 5 (29.4%) were grouped as CTX intermediate. In the case of AMC, 15 strains (88.2%) were resistant and 2 (11.8%) were intermediate. All tested strains were susceptible to ERY. Generally, the vast majority of strains were also susceptible to CIP and GEN, but one strain was resistant to both of them. In the case of DOX, 13 strains (76.5%) were classified as intermediate. Resistance to SXT was observed for six strains (35.3%), while one strain (5.9%) was intermediate, and ten strains (58.8%) were susceptible to this antimicrobial. Resistance phenotypes with three or four antimicrobial agents were observed, including PEN/AMC/CTX/SXT (S107, S116, S124 and S175), P/AMC/CTX (S81, S83, S87, S125, S128, S132 and S170), PEN/AMC/SXT (S29), PEN/CTX/SXT (S11) and PEN/CIP/GEN (S177). Strains isolated from the same herds did not present the same resistance profiles.

### 2.3. Pulsed-Field Gel Electrophoresis Analysis

#### 2.3.1. *S. aureus*

PFGE analysis showed three different pulsotypes among five *S. aureus* isolates. One pulsotype grouped three strains, while two strains showed a specific pulsotype. One cluster was distinguished that grouped four strains with a similarity index of 84.2% (Figure 1). Each of the animals from which *S. aureus* strains were isolated represented a different herd.

#### 2.3.2. *Streptococcus* spp.

All 48 isolates of *Streptococcus* spp. were analyzed by PFGE. Three strains of *S. equi* subsp. *zooepidemicus* that were obtained from different herds showed the same pulsotype. On the other hand, broad genetic diversity was obtained for the remaining 45 *S. dysgalactiae* strains isolated from 42 animals. The dendrogram analysis showed seven clusters with similarity indexes ranging from 85.7% to 100% after *SmaI* DNA digestion (Figure 2). These clusters grouped two strains (clusters 1, 4 and 7), three strains (clusters 2 and 3), four strains (cluster 6) and thirteen strains (cluster 6). Clusters 2, 5 and 6 included some strains from the same herds. A total of 25 pulsotypes were observed among the tested *S. dysgalactiae* strains. The most common pulsotype was characteristic of twelve strains. One pulsotype grouped four strains, one grouped three strains, and four grouped two strains each. Unique pulsotypes were observed for 18 strains. In addition, the PFGE analysis showed that genetically different *S. dysgalactiae* strains circulated within a single herd, while genetically similar or identical strains occurred in different herds.

In one case, two different pulsotypes were obtained from different organs of the same animal with a similarity of 36.3%. In another case, three pulsotypes were obtained from the same animal from different organs, two of which (S30 and S31) belonged to the same cluster: the first two had a similarity of 100%, and the third (S27) had 58.7% similarity.

The evaluation of the dendrogram showed no significant difference between isolates from lesioned and non-lesioned lymph nodes. For isolates from lesioned lymph nodes (*n* = 31), we obtained 19 pulsotypes and distinguished five clusters, while for isolates from non-lesioned lymph nodes (*n* = 14), we obtained 11 pulsotypes and distinguished four clusters.

#### 2.3.3. *R. equi*

The PFGE analysis showed high genetic diversity among 17 *R. equi* isolates. Eleven different pulsotypes were obtained. Pulsotype 6.2 was the most prevalent as it was found in five isolates, followed by pulsotypes 6.3 and 7, which were found in two isolates each. Eight *R. equi* strains each presented a unique pulsotype (Figure 3). Both same and different pulsotypes were present in animals from the same farm. Two clusters that grouped nine and two strains with 81.3% and 100% similarity, respectively, were distinguished.

## 3. Discussion

In this research, we evaluated the genetic relatedness and antimicrobial resistance of bacteria isolated from pig lymph nodes collected during post-mortem examination at a slaughterhouse. One of the main limitations of this study was the restriction of the geographic area of the study to Poland as the pigs came from only three provinces: Wielkopolska (Greater Poland), Łódzkie and Kujawsko-Pomorskie. Another limitation was the freezing of lymph nodes prior to testing, which may have resulted in the reduced growth of some bacterial species.

The farm types and herd sizes are typical of Poland [29]. In our study, we isolated bacteria from lymph nodes both with and without lesions. Pathogens isolated from lymph nodes in pigs may be zoonotic, including *S. aureus* [30], *S. dysgalactiae* [31,32], *S. equi* subsp. *zooepidemicus* [11], *R. equi* [33,34] and *T. pyogenes* [35]. Not only pork consumers but also abattoir workers, veterinarians and other industry workers who come into direct contact with infected tissues are particularly vulnerable to infection from these microorganisms.

The microbiological analysis showed that 48 isolates belonged to *Streptococcus* spp., including 45 *S. dysgalactiae* and 3 *S. equi* subsp. *zooepidemicus*; 17 isolates were *R. equi*; and 5 were *S. aureus*. The obtained results are similar to previously published reports regarding isolated pathogens, but the frequency of the isolation of each species varied [1,5,13,17,36]. In a study conducted by Lara et al. (2011), *R. equi* and *Streptococcus* spp. were the predominant species isolated from porcine lymph nodes, similarly to in our study. Generally, *Streptococcus* spp. were often isolated as one of the main pathogens from purulent lesions of lymph nodes in pigs [1,5,17]. On the other hand, it should be mentioned that the frequency of *T. pyogenes* isolation in this study was much lower than previously reported in pigs [1,5].

However, compared to previous reports, significantly more pathogens were isolated from lymph nodes without lesions in our study [14]. In two cases, both *S. dysgalactiae* and *R. equi* were isolated from the same lymph node without lesions. The isolation of a few kinds of bacteria from one lymph node sample has previously been reported [17].

The most common bacterium isolated in this study was *S. dysgalactiae*. However, the identification of *S. dysgalactiae* strains into subspecies was not performed because the taxonomy of *S. dysgalactiae* remains unclear. Recent studies indicate that *S. dysgalactiae* may be segregated into two distinct groups: one includes only human strains, while the other includes animal strains of swine, canine and equine origin, as well as more phylogenetically distinct bovine strains [37]. Most *S. dysgalactiae* isolates in this study belonged to Lancefield group C, as is consistent with previous reports [7,38], but some isolates belonged to group A. To date, there are no reports in the literature of infections caused by *S. dysgalactiae* from Lancefield group A in pigs. Moreover, it should be highlighted that the presence of group A *S. dysgalactiae* strains are very rare. It is worth noting that *S. dysgalactiae* group A has zoonotic potential and has been reported in patients with bacteremia [39].

To assess genetic relatedness, we used the PFGE method, which has been recognized as the “gold standard” due to its excellent discriminatory power in studies of various groups of bacteria [23,40]. High genetic variation was observed for *S. dysgalactiae* strains of 25 different PFGE pulsotypes. These results are consistent with previous studies on the genetic diversity of streptococci from animals, including *S. dysgalactiae* [5,9]. The PFGE method used showed a high degree of intra-species polymorphism, both between isolates from different hosts and between isolates from the same host. As in the study by Cardoso-Tosset et al. (2020) [5], the PFGE analysis showed that genetically different *S. dysgalactiae* isolates circulated within one herd, while genetically similar isolates circulated in different herds. The significant polymorphism revealed by the PFGE method among *S. dysgalactiae* strains confirms that it is a highly discriminating tool in epidemiological studies of animal infections caused by this bacterium.

Low diversity in PFGE patterns was observed for the *S. equi* subsp. *zooepidemicus* and *S. aureus* strains; however, it should be noted that the number of tested strains was limited. Three *S. equi* subsp. *zooepidemicus* strains from different farms were of a single pulsotype. This is in line with the results of genotyping *S. equi* subsp. *zooepidemicus* isolates from pigs and monkeys in Indonesia [12].

In the case of *S. aureus*, three pulsotypes were obtained for five strains from different herds, and four strains belonged to one cluster. There are limited data in the literature on the PFGE typing of *S. aureus* strains isolated from pigs [41,42]. In our study, PFGE patterns showed high similarity. Comparable results were obtained in a study by Broens (2011), which found that MRSA-negative pigs can be colonized by MRSA-positive strains during transport from farm to slaughterhouse, suggesting that the source of MRSA colonization in pigs could be contaminated trucks, slaughterhouses or staff. In addition, mixing pigs from different farms in transport and in slaughterhouse livestock stores could also lead to colonization by MRSA strains [41,43]. However, there are also reports indicating wide variation in PFGE patterns between MRSA strains isolated from pigs, thus suggesting extensive DNA rearrangements and the potential for the gain or loss of genetic traits, i.e., genomic events occurring over short timescales [44].

Genotyping *R. equi* strains by PFGE showed high genetic diversity between the studied isolates. Two isolates (S170 and S181) from the same farm (AB) showed the same pulsotype; however, the same pulsotype was also found in isolates (S125, S132 and S177) from pigs from different farms (AE, AR and AS). Similar results were obtained by Pate et al., who evaluated the genetic diversity of *R. equi* isolated from pigs in Slovenia [45].

In our study, the strip diffusion method, which is widely applied in diagnostic laboratories, was used to assess antimicrobial susceptibility. The studied streptococcal strains showed the lowest MIC values for β-lactam antibiotics, aminoglycosides, fluoroquinolones and sulfonamides, while tetracyclines and macrolides presented the highest MIC values. There are only a few reports on the antimicrobial susceptibility of *Streptococcus* spp. isolates from pigs. The results of our study confirmed that β-lactam antibiotics are the best choice for treatment, which is consistent with previous studies in pigs and humans [7,46]. In our study, macrolides and tetracyclines showed the highest MIC50 and MIC90 values among the dilution ranges tested, which is consistent with antibiotic resistance studies on *S. dysgalactiae* strains from pigs from Brazil, Austria and Korea [7,46,47]. It is noteworthy that macrolides and tetracyclines are commonly used to treat infections in livestock, and in our study, resistance to these classes of antibiotics was found in strains isolated from lymph nodes both with and without lesions. Interestingly, a study by Korean researchers on the antimicrobial susceptibility of human *S. dysgalactiae* strains showed a marked increase in the prevalence of erythromycin resistance: from 9.4% to 34.8% [48]. The results of our study appear to be an important public health warning as people with tonsillopharyngitis due to streptococcal infection who are allergic to β-lactam antibiotics are usually treated with macrolides as an alternative antimicrobial therapy. If macrolide resistance is confirmed, then tetracyclines and fluoroquinolones should be considered as a second choice [7,49].

Moreover, in this research, the occurrence of macrolide and tetracycline resistance genes among *S. dysgalactiae* strains was investigated. There are few reports of antimicrobial resistance in *S. dysgalactiae* swine strains. In comparison to data from a Korean study published in 2020, we obtained the same results regarding the lack of the *ermA* gene in the *S. dysgalactiae* strains studied [7]. In contrast, the *ermB* gene in this study was present in 7.8% of the strains tested, while in the study by Sang Oh et al., 18.2% of the strains carried this gene [7]. Among tetracycline resistance genes, our results differed from previous studies. In our study, the *tetM* gene was more dominant (42.2%) compared to the *tetO* gene (6.8%); in isolates from Korea, *tetO* was predominant (90.9%), and *tetM* was carried by 18.2% of strains [7]. However, similarly to our results, the *tetK* and *tetL* genes were not detected [7]. There is a noticeable discrepancy between the phenotypic and genotypic antimicrobial resistance of *S. dysgalactiae* isolates. This can be explained by the observation, as reported by Smith M. et al., that the bacterial resistance phenotype possessed is not always an accurate reflection of antimicrobial resistance genes [7,50]. In addition, the presence in the tested strains of other genes that were not detected in this study cannot be ruled out.

All tested *S. aureus* isolates showed susceptibility to some β-lactam antibiotics, aminoglycosides, macrolides, tetracyclines and sulfonamides. Of the five strains, two were resistant (40%), and three had an intermediate susceptibility (60%) to CTX. Comparing our results to previous studies, we did not find resistance to tetracyclines, penicillins, macrolides, or lincosamides, whereas resistance to all four groups was found among the *S. aureus* strains isolated from pigs at farmers’ markets in the US [51]. The strains we tested were significantly more susceptible to antibiotics compared to the strains described in a Belgian swine survey in 2013, in which 98.6% of strains showed resistance to tertacycline, 96.2% to trimethoprim, 61.1% to ciprofloxacin, 57.8% to erythromycin and 45.5% to gentamicin [52]. In addition, these authors showed a significant increase in resistance to ciprofloxacin, from 32% to 61.1% year-on-year [52].

All tested *R. equi* strains were shown to be susceptible to macrolides, and the vast majority of strains were also susceptible to fluoroquinolones and aminoglycosides, which is consistent with data from the literature [53]. There are no publications that provide data on drug resistance in *R. equi* from pigs, making our findings impossible to compare. However, a high susceptibility to aminoglycosides and tetracyclines and a relative susceptibility to macrolides were noted for isolates obtained from horses [54]. The presence of *R. equi* strains classified as moderately sensitive to tetracyclines in our study may be related to the method we used because some research has suggested that the strip diffusion method may underestimate MIC values for this class of antibiotics [54]. Thus, it can be presumed that the tested strains were resistant to tetracyclines. The high resistance of *R. equi* strains to β-lactam antibiotics observed in our study was not surprising since this is not uncommon [55].

In Poland, pork is the most frequently consumed type of meat [56]; thus, it seems important to know the microbiological factors present in the lymph nodes. The characteristics of isolates, including the assessment of their drug resistance, makes it possible to determine their zoonotic potential and whether they pose a potential threat to consumer health. This is particularly important considering that pork sausages that are not subject to thermal processing are produced in Poland and many regions of the world.

Research into the prevalence of *Mycobacteriaceae* in swine lymph nodes has been described previously [2].

## 4. Materials and Methods

### 4.1. Material Collection

During post-mortem inspection in slaughterhouses, mandibular lymph nodes were collected from 199 pigs. (In two cases, two samples were collected from 1 individual pig. In one case, seven samples were collected from 1 individual pig as general lesions were found throughout the carcass. Tracheobronchial, mediastinal, first rib, mesenteric and porta hepatis lymph nodes were collected for examination. Lung and liver organ samples were also collected.) Lymph nodes, organs with lesions (*n* = 95) and randomly selected unchanged mandibular lymph nodes (*n* = 113) were then tested for the presence of the selected pyogenic bacteria. The material for the research was collected in two slaughterhouses in the Greater Poland region, while the animals came from 83 farms located in central Poland, with an average number of 117 individuals in a herd. The mean age of the pigs was 6.4 months. Of the 199 pigs tested, 99 were male and 100 were female. The collected material was stored in a freezer at −20 °C before analysis.

### 4.2. Bacterial Isolation and Identification

After thawing, the tissues were minced with sterile scissors and placed into bags with a BagPack (BagPage^®^ 100) filtering membrane. Then, sterile 0.9% saline solution was added and homogenized using a stomacher at 12 strokes/1 s. The resulting suspension was poured into 15 mL tubes and centrifuged at 1500× *g*. In the next step, the supernatant was poured out, and the pellet was used for further analysis.

After homogenization, the supernatant was transferred to Columbia Agar supplemented with 5% sheep blood (CAB) (Graso Biotech, Starogard Gdański, Poland) using sterile cotton swabs. It was then smeared and cultured for 48 h at 37 °C under microaerophilic conditions. Isolates of the tested bacterial species were primarily identified based on their growth features, including the type of hemolysis on CAB; the cell morphology, including Gram staining; and the catalase activity. Moreover, for the identification of *R. equi*, the CAMP test with *S. aureus* ATCC^®^25923 reference strain was performed. The result of the CAMP test was read after 48 h of incubation at 37 °C under aerobic conditions. The Microgen^®^Staph latex agglutination tests (Graso Biotech, Starogard Gdański, Poland) were performed to identify *S. aureus* isolates. The identification of *S. aureus* isolates was confirmed using the PCR technique, as described previously [57]. *Streptococcus* spp. isolates were identified at the species level based on a PCR assay using *Sdy519* and *Sdy920* primers designed for the highly divergent and species-specific region of the gene encoding 16S rRNA of *S. dysgalactiae* [58] or a PCR assay with the primer sets *SodA-F* and *SodA-R* for the *sodA* gene of *S. equi* [59]. The differentiation of *S. equi* subsp. *zooepidemicus* and *S. equi* subsp. *equi* was performed by detecting the seeI gene encoding superantigenic toxin of *S. equi* subsp. *equi*, as described by Alber et al. (2004). Additionally, the Microgen^®^Strep latex agglutination test (Graso Biotech, Starogard Gdański, Poland) was used to serotype the streptococci isolates. All studied isolates were stored at −20 °C in a tryptic soy broth (TSB) containing 20% glycerol (*v*/*v*) for future use.

### 4.3. DNA Extraction

The boiling method was used to extract DNA from all tested isolates. Colonies grown on CAB were suspended in 500 µL of nuclease-free water. The suspension was heated at 99 °C for 10 min, after which it was cooled on ice and centrifuged (6 min, 8000× *g*). The supernatant was collected, stored at −20 °C and used for further PCR studies.

### 4.4. Antimicrobial Susceptibility Testing

Using a strip diffusion method with Liofilchem^®^MIC Test Strips (Liofilchem, Via Scozia, Italy), the MIC values for eight antimicrobials were determined for the tested isolates: amoxicillin with clavulanic acid (AMC; 0.016–256 μg/mL), penicillin (PEN; 0.002–32 μg/mL), cefotaxime (CTX; 0.002–32 μg/mL), erythromycin (ERY; 0.016–256 μg/mL), doxycycline (DOX; 0.016–256 μg/mL), gentamicin (GEN; 0.016–256 μg/mL), ciprofloxacin (CIP; 0.002–32 μg/mL) and trimethoprim-sulfamethoxazole (SXT; 0.002–32 μg/mL). The bacterial suspension in saline (a density of 0.5 McFarland standard) was inoculated on Mueller–Hinton Agar supplemented with 5% sheep blood (Graso Biotech, Starogard Gdański, Poland). The plate was covered with strips and was incubated for 24 h at 37 °C under aerobic conditions. The testing conditions used in the study were in accordance with the CLSI guidelines [27] for the studied bacterial species. The MIC value was read at the point where the edge of the growth inhibition ellipse intersected the strip. Moreover, the MIC50 (antibiotic concentration required to inhibit the growth of 50% of isolates) and MIC90 (antibiotic concentration required to inhibit the growth of 90% of isolates) values were determined for each antimicrobial agent. Three reference strains, *S. dysgalactiae* subsp. *equisimilis* ATCC^®^12394, *S. aureus* ATCC^®^25923 and *R. equi* ATCC^®^6939, were also included in the study for quality control.

The MIC breakpoints used in this study are shown in Table 2. In the case of *Streptococcus* spp., the MIC breakpoints for PEN, CIP (as for enrofloxacin), ERY, DOX (as for tetracycline), GEN and SXT were based on the criteria recommended for *Streptococcus* spp., as defined by the Antibiogram Committee of the French Microbiology Society (CA-SFM) guidelines VET2021 [26]. For CTX, the criterium for cefpodoxime specific to *Streptococcus canis* according to the Clinical Laboratory Standards Institute (CLSI) guidelines VET08 was adopted [27]. Moreover, the criterium for *Streptococcus* spp. isolates from cats against AMC was also adapted from CLSI VET08 [27]. For *S. aureus*, the criteria for PEN, CIP (for enrofloxacin), ERY, DOX (for tetracycline), GEN and SXT were as defined for *Staphylococcus* spp. in the CA-FSM guidelines VET2021 [26]. The MIC breakpoint for CTX was the same as that for cefpodoxime in the CLSI VET08 document for *S. aureus* and *Staphylococcus pseudintermedius* in the case of dogs, and AMC was also adapted from the same guidelines as criteria for *Streptococcus* spp. categorization of canine isolates [28]. For *R. equi*, ERY and DOX were interpreted according to the criteria adopted for *R. equi* in CLSI document VET06 [29], while other antimicrobials were referenced to *Staphylococcus* spp. values according to the CA-FSM guidelines VET2021 and CLSI VET08 [26,27].

### 4.5. Detection of Antimicrobial Resistance Genes

The presence of genes commonly associated with erythromycin resistance (*ermA* and *ermB*) and tetracycline resistance (*tetK/L*, *tetM* and *tetO*) was assessed for the streptococci strains using PCR with the primers and conditions presented in Table 3. Universal primers that detect tetracycline resistance genes encoding ribosome protection proteins were used first. Then, in the case of the isolates that tested positive in this reaction, the specific primers for *tetM* and *tetO* were used. All PCR mixtures contained 1 µL of each primer (10 pmol/µL), 12.5 µL of DreamTaq Green PCR Master Mix (2×) (Thermo Fisher Scientific, Waltham, MA, USA), 40 ng of DNA template and nuclease-free water up to 25 µL. Reaction products were recognized by electrophoresis in 1% (*w*/*v*) agarose gel in Tris-Acetate-EDTA (TAE) buffer with Midori Green DNA Stain (Nippon Genetics, Düren, Germany), visualized and analyzed using a Gel DocTM EZ Imaging System with Image Lab Software (version 5.2.1) (Bio-Rad, Hercules, CA, USA).

### 4.6. Pulsed-Field Gel Electrophoresis

#### 4.6.1. Analysis of *S. aureus* Isolates

The pulsed-field gel electrophoresis (PFGE) procedure was adapted from previous research [65,66] with some modifications. The *S. aureus* isolates collected from a 24 h culture on CAB were suspended in saline to achieve a density of 3.5 on the McFarland standard. The bacterial suspension was mixed with 2% CleanCut Agarose (Bio-Rad, Hercules, CA, USA). The obtained agarose discs were incubated for 18 h at 37 °C in a lysis solution with 2 mg/mL of lysozyme (Sigma-Aldrich, Steinheim am Albuch, Baden-Württemberg, Germany), 5 µg/mL of RNase (A&A Biotechnology, Gdańsk, Poland) and 50 µg/mL of lysostaphin (A&A, Gdańsk, Biotechnology). After the indicated time, the discs were transferred to the solution with 1 mg/mL of proteinase K (A&A Biotechnology, Gdańsk, Poland), where they were incubated for 24 h at 50 °C. The agarose discs were digested with *SmaI* (20 U/µL) (Thermo Fisher Scientific, Waltham, MA, USA) overnight at 25 °C. The restriction fragments were separated in 1.2% agarose gel (*w*/*v*). The separation program was a running time of 20, a temperature of 14 °C, a voltage gradient of 6 V/cm, an initial pulse time of 5 s and a final pulse time of 30 s. The reference strain S. aureus ATCC^®^25923 was also used in this study.

#### 4.6.2. Analysis of *Streptococcus* spp. Isolates

PFGE was performed as described by Vela et al. (2003) [24] with some changes. Briefly, the overnight cultures of *Streptococcus* spp. isolates on CAB were suspended in saline to obtain a density of 4 according to the McFarland standard. An equal volume of bacterial suspension and 2% CleanCut Agarose (Bio-Rad, Hercules, CA, USA) were mixed to prepare agarose discs. The agarose discs were incubated overnight at 37 °C with 1 mg/mL of lysozyme (Sigma-Aldrich, Steinheim am Albuch, Baden-Württemberg, Germany) and then incubated for another 24 h at 50 °C with 500 μg/mL of proteinase K (A&A Biotechnology, Gdańsk, Poland). To perform the restriction digest, a solution with 20 U/μL of *SmaI* (Thermo Fisher Scientific, Waltham, MA, USA) was prepared and incubated overnight at 25 °C. A total of 1.1% (*w*/*v*) agarose gel was used for the electrophoresis with the following parameters: a running time of 21 h, a temperature of 14 °C, a voltage gradient of 6 V/cm, an initial pulse time of 1 s and a final pulse time of 30 s. The reference strain *S. dysgalactiae* subsp. *equisimilis* ATCC^®^12394 was also used in this study.

#### 4.6.3. Analysis of *R. equi* Isolates

PFGE was performed as previously described [67,68] with minor modifications. Briefly, overnight cultures of *R. equi* in brain–heart infusion (BHI) broth with 1% glycerol, 0.4% glucose and 0.2% TWEEN^®^85 were adjusted to OD600 0.65, and the cells were incorporated into 1.5 % (*w*/*v*) agarose discs using Top Vision Low Melting Point Agarose (Thermo Fisher Scientific, Waltham, MA, USA). After 18 h lysis with 20 mg/mL of lysozyme (Sigma-Aldrich, Germany) and 50 μg/mL of RNase (A&A Biotechnology, Gdańsk, Poland) at 37 °C, the discs were incubated overnight with 20 mg/mL of proteinase K (A&A Biotechnology, Gdańsk, Poland) at 50 °C. Then, the agarose discs containing DNA were digested with 10 U/μL of *VspI* (Thermo Fisher Scientific, Waltham, MA, USA) overnight at 37 °C. For the electrophoresis, 1.1% (*w*/*v*) agarose gel was used with running conditions as follows: a running time of 22 h, a temperature of 14 °C and a voltage gradient of 6 V/cm. The program ran twice: during the first run (7 h), an initial pulse time of 6 s and a final pulse time of 15 s were used; during the second run (15 h), an initial pulse time of 23 s and a final pulse time of 40 s were used. The reference strain, *R. equi* ATCC^®^6939, was also used in this study.

The restriction fragments of the tested isolates were separated by clamped homogenous electric field electrophoresis with a CHEF-DR II System (Bio-Rad, Hercules, CA, USA) in a 0.5× TBE buffer. The gels were stained with ethidium bromide for 15 min, destained in distilled water and visualized and photographed using a Gel DocTM EZ Imaging System with Image Lab Software (version 5.2.1) (Bio-Rad, Hercules, CA, USA). BioNumerics software version 7.6 (Applied Maths, Sint-Martens-Latem, Belgium) was used for the PFGE results analysis. The cluster analysis was performed by the unweighted pair group method with arithmetic mean (UPGMA) using the Dice similarity coefficient with optimization and position tolerance set at 1%. The strains were clustered using an 80% homology cut-off, above which strains were assigned to the same cluster because they were considered to be closely related [69].

## 5. Conclusions

In summary, in this study, the most common pyogenic bacteria isolated from swine lymph nodes, including lymph nodes without lesions, were *S. dysgalactiae*, *R. equi* and *S. aureus*. The occurrence of pathogenic bacteria in lymph nodes without lesions is alarming because this prevents pathogen detection during meat inspection at slaughterhouses. In addition, it should be noted that some bacterial strains showed antimicrobial resistance and carried antibiotic-resistant genes. Thus, the presence of these potentially zoonotic pathogens should be considered when testing meat from slaughtered pigs. Moreover, pork can be a potential source of bacteria, thus causing a significant risk to human health. Therefore, it seems reasonable to monitor the occurrence of these bacteria in pig herds.

In addition, it should be highlighted that bacterial strains with the same PFGE profile occurred in pigs from the same herd, suggesting the possibility of pathogen transmission between animals.

## Figures and Tables

**Figure 1 antibiotics-12-01026-f001:**
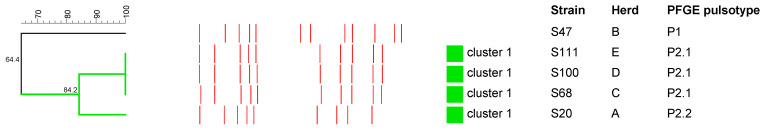
Dendrogram showing the degree of similarity among the five tested *S. aureus* strains based on the results of PFGE analysis. Red lines show the obtained PFGE patterns. One cluster (green square) was defined from groups of closely related strains sharing at least 80% similarity.

**Figure 2 antibiotics-12-01026-f002:**
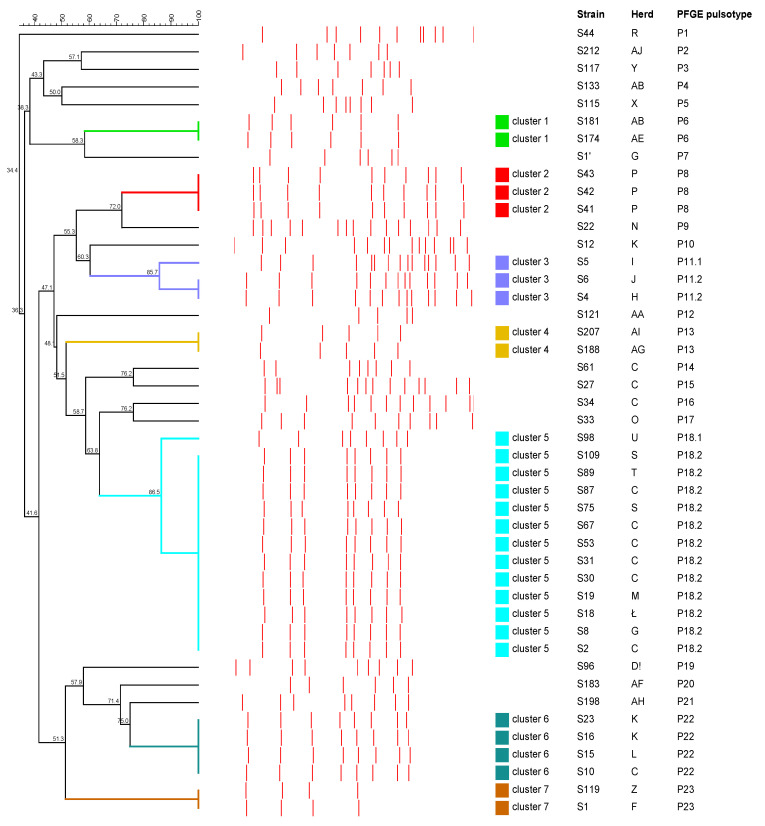
Dendrogram showing the degree of similarity among 45 tested *S. dysgalactiae* strains based on the results of PFGE analysis. Red lines show the obtained PFGE pulsotypes. Seven clusters (colored squares) were defined from groups of closely related strains sharing at least 80% similarity.

**Figure 3 antibiotics-12-01026-f003:**
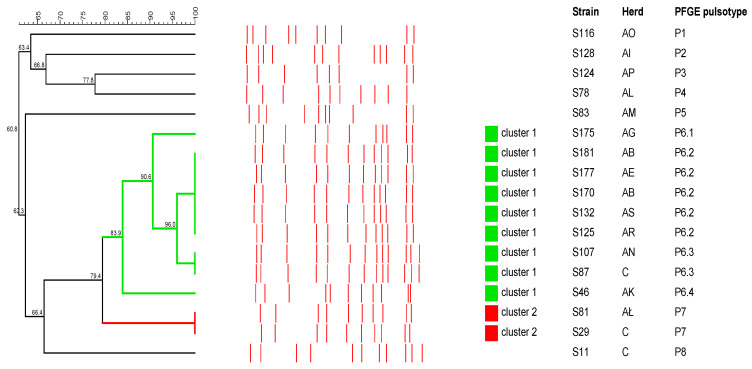
Dendrogram showing the degree of similarity among the tested *R. equi* strains based on the PFGE typing. Red lines show the obtained PFGE patterns. Two clusters (colored squares) were defined from groups of closely related strains sharing at least 80% similarity.

**Table 1 antibiotics-12-01026-t001:** The result of microbiological examination of the lymph nodes and tissues collected from pigs in this study (total number of isolates = 70).

Samples	*Streptococcus dysgalactiae*	*Streptococcus equi* subsp. *zooepidemicus*	*Staphylococcus aureus*	*Rhodococcus equi*
L.n. mandibular with lesions (*n* = 86)	28	0	4	8
L.n. mandibular without lesions (*n* = 113)	14	3	1	9
L.n. mediastinal with lesions (*n* = 1)	1	0	0	0
L.n. of the hilum of a liver with lesions (*n* = 1)	1	0	0	0
Liver with lesions (*n* = 1)	1	0	0	0
L.n. of the first rib with lesions (*n* = 1)	0	0	0	0
L.n. retropharyngeal with lesions (*n* = 1)	0	0	0	0
L.n. tracheobronchial with lesions (*n* = 2)	0	0	0	0
L.n. mesenteric with lesions (*n* = 1)	0	0	0	0
Lung with lesions (*n* = 1)	0	0	0	0
**Total**	**45**	**3**	**5**	**17**

L.n.—Lymph node.

**Table 2 antibiotics-12-01026-t002:** Minimal inhibitory concentrations of 50 and 90 of selected antimicrobials for *Streptococcus spp*. isolates (*n* = 48), *R. equi* isolates (*n* = 17) and *S. aureus* isolates (*n* = 5).

	Antimicrobial Agent ^a^	MIC (μg/mL)		MIC Breakpoints (μg/mL) ^b^	
		MIC_50_	MIC_90_	S	R
***Streptococcus* spp.**	**PEN**	0.0006	0.016	≤0.25	>16 ^c^
	**AMC**	0.016	0.047	≤0.25/0.12	≥1/0.5 ^d^
	**CTX**	0.032	0.094	≤2	≥8 ^d^
	**CIP**	0.5	0.75	≤0.5	>2 ^c^
	**ERY**	0.19	0.75	≤1	>4 ^c^
	**DOX**	0.5	24	≤4	>8 ^c^
	**GEN**	1.5	3	≤250	>500 ^c^
	**SXT**	0.38	3	≤2/38	>8/152 ^c^
** *Rhodococcus equi* **	**PEN**	12	>32	≤0.25	>0.25 ^c^
	**AMC**	1.5	2	≤0.25/0.12	≥1/0.5 ^d^
	**CTX**	>32	>32	≤2	≥8 ^d^
	**CIP**	0.75	1.5	≤2	>2 ^c^
	**ERY**	0.25	0.38	≤0.5	≥8 ^e^
	**DOX**	8	12	≤4	≥16 ^e^
	**GEN**	0.5	0.75	≤1	>1 ^c^
	**SXT**	2	>32	≤2/38	>8/152 ^c^
** *Staphylococcus aureus* **	**PEN**	0.032	0.125	≤0.25	>0.25 ^c^
	**AMC**	0.125	0.25	≤0.25/0.12	≥1/0.5 ^d^
	**CTX**	4	>32	≤2	≥8 ^d^
	**CIP**	0.25	0.75	≤2	>2 ^c^
	**ERY**	0.094	0.5	≤1	>4 ^c^
	**DOX**	0.38	0.5	≤4	>8 ^c^
	**GEN**	0.19	0.19	≤1	>1 ^c^
	**SXT**	0.19	0.38	≤2/38	>8/152 ^c^

^a^ PEN—Penicillin; AMC—amoxicillin/clavulanic acid; CTX—cefotaxime; CIP—ciprofloxacin; ERY—erythromycin; DOX—doxycycline; GEN—gentamicin; SXT—trimethoprim-sulfamethoxazole; ^b^ S—susceptible; R—resistant. ^c^ According to the CA-SFM document VET2021 [26]; ^d^ according to the CLSI document VET08 [27]; ^e^ according to the CLSI document VET06 [28].

**Table 3 antibiotics-12-01026-t003:** Primers and PCR conditions used in this study.

Primer Designation	Primer Sequence (5′-3′)	Target Gene	Annealing Temperature (°C)	Amplicon Size (bp)	Reference
**DI_F**	GAYACICCIGGICAYRTIGAYTT	*Tet* ^a^	53 ^b^	1100	[60]
**DII_R**	GCCCARWAIGGRTTIGGIGGIACYTC				
**tetM_F**	TTAAATAGTGTTCTTGGAG	*tetM*	54 ^c^	656	[61]
**tetM_R**	CTAAGATATGGCTCTAACAA				
**tetO_F**	GGCGTTTTGTTTATGTGCG	*tetO*	50 ^c^	559	[62]
**tetO_R**	ATGGACAACCCGACAGAAGC				
**TKI_F**	CCTGTTCCCTCTGATAAA	*tetK/L*	50 ^b^	1050	[63]
**TL32_R**	CAAACTGGGTGAACACAG				
**ermA_F**	TCTAAAAAGCATGTAAAAGAA	*ermA*	52 ^c^	645	[64]
**ermA_R**	CTTCGATAGTTTATTAATATTAGT				
**ermB_F**	GAAAAGGTACTCAACCAAATA	*ermB*	52 ^c^	639	[64]
**ermB_R**	AGTAACGGTACTTAAATTGTTTAC				

^a^ Universal primers that detect the tetracycline resistance genes that encode ribosome protection proteins. ^b^ PCR conditions: initial denaturation at 95 °C for 3 min; 35 cycles of denaturation at 95 °C for 1 min; annealing for 1 min at variable temperatures; extension at 72 °C for 2 min; final extension at 72 °C for 5 min. ^c^ PCR conditions: initial denaturation at 95 °C for 3 min; 30 cycles of denaturation at 95 °C for 45 s; annealing for 45 s at variable temperatures; extension at 72 °C for 1 min; final extension at 72 °C for 2 min.

## Data Availability

Not applicable.

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
