# Peer review of "The Genetic Diversity and Antimicrobial Resistance of Pyogenic Pathogens Isolated from Porcine Lymph Nodes"

_antibiotics, 2023, doi:10.3390/antibiotics12061026_

Round 1

Reviewer 1 Report

The manuscript is interesting and well structured. Some minor grammar and syntax errors that have been spotted, not serious enough to hinder the understanding of it, can be easily cured by the revision of the text by a native speaker. What is of concern though is the small number of isolates collected, especially in the case of Staphylococcus aureus. This can be explained by the methodological flaw of freezing the lymph nodes after collection. This does not permit the comparison of the results (e.g. prevalence etc.) with the results of other researchers. Still the research on the isolates collected can be  of interest.

P2 L57. Since the new taxonomy has revised the name of R. equi, why do the authors use this instead of Prescottella equi? Perhaps it would be more appropriate to report it as P. equi reporting the older R. equi instead.

P2 L94 to P3 L116. In terms of the lymph nodes freezing, is it appropriate to report the percentage of samples tested positive?

P4 L130. Since the abbreviations MIC50 and MIC90 are recorded elsewhere there is no need for repetition. Therefore, add the full name or the abbreviation but not both.

P4 L136 and elsewhere. In some cases, the italics of scientific names of the bacteria examined have not been retained. Please revise.

P9, L259-272. Although interesting, the authors have not worked on the taxonomy of Streptococcus dysgalactiae. Therefore, this paragraph section seems not relevant.

P11 L382-393. The lessions observed could be of interest. Also a map showing the position of the abattoirs and the area of the farms from which the pigs originated could be of interest.

P11 L396-8. Is there a reference for this methodology? Which was the purpose of the centrifugation?

P12 L426- I am wondering if the use of the veterinary breakpoints are appropriate. The authors have collected these isolates as potential pathogens to humans. Perhaps the relevant CLSI Supplements targeting the human isolates are more appropriate to interpret the results. Please comment.

P12 L439-441. I am wondering which was the purpose of the calculation of MIC50 and MIC90. Usually, they are calculated in large surveys for the establishment of breakpoints. In similar papers, the characterization of the isolates as resistant or not is usually made. Please comment.  

P14 L540. Since there was this methodological flaw, I would not consider this appropriate.

The manuscript has a methodological flaw, that of freezing samples prior to examination. This is not prohibitive of further evaluation of results, but the authors should exercise extreme care in their comparisons since they can be hindered by this flaw. For example is not possible to report that these were the most common pathogens since other bacteria causing of purulent lessions could have been killed during freezing, nor can the authors compare prevalence to other authors.

Author Response

Comment: This can be explained by the methodological flaw of freezing the lymph nodes after collection. This does not permit the comparison of the results (e.g. prevalence etc.) with the results of other researchers.

Answer: Thank you for your comment. We are aware that this is a limitation of our study, and it has been highlighted in the article (Page 9, Line 237-239). Freezing of the lymph nodes was dictated by the difficulties associated with the collection of the material. We have analyzed the references once again in the context of the type of studied specimens and one new reference has been included (Page 9, Line 249-251)

Comment: P2 L57. Since the new taxonomy has revised the name of R. equi, why do the authors use this instead of Prescottella equi? Perhaps it would be more appropriate to report it as P. equi reporting the older R. equi instead.

Answer: Thank you for your comment. The taxonomy of this species is not clearly established. The name Prescottella equi is a synonym (validly published under the International Code of Nomenclature of Prokaryotes) and Rhodococcus equi is a correct name; however, this bacterium is also known as Rhodococcus hoagie. In our work, we use the well-known name R. equi, but the name Prescottella equi has been also mentioned  (Page 1, Line 17; Page 1, Line 42)

Comment: P2 L94 to P3 L116. In terms of the lymph nodes freezing, is it appropriate to report the percentage of samples tested positive?

Answer: Thank you for your question. The frequency of isolation of bacteria was given in the results as a prevalence, however  the limitations were clearly indicated in the Discussion.

Comment: P4 L130. Since the abbreviations MIC50 and MIC90 are recorded elsewhere there is no need for repetition. Therefore, add the full name or the abbreviation but not both.

Answer: Thank you. We have corrected this mistake. (Page 2, Line 131-132)

Comment: P4 L136 and elsewhere. In some cases, the italics of scientific names of the bacteria examined have not been retained. Please revise.

Answer: Thank you. We have corrected this mistake.

Comment: P9, L259-272. Although interesting, the authors have not worked on the taxonomy of Streptococcus dysgalactiae. Therefore, this paragraph section seems not relevant.

Answer: We agree with this comment, this paragraph has been rewritten. (Page 10, Line 263-270)

Comment: P11 L382-393. The lessions observed could be of interest. Also a map showing the position of the abattoirs and the area of the farms from which the pigs originated could be of interest.

Answer: Thank you for your comment. We have added a map, however, in order to make it readable, we have marked on the map only the location of the farms from which the purulent bacteria were isolated from animals. (P17, Line 576-592, Figure S.1)

Comment: P11 L396-8. Is there a reference for this methodology? Which was the purpose of the centrifugation? 

Answer: Thank you for your comment. We have corrected the mistake, of course, we were transferred to Columbia Agar the pellet, not supernatant. (Page 12, Line 405-406)

Comment: P12 L426- I am wondering if the use of the veterinary breakpoints are appropriate. The authors have collected these isolates as potential pathogens to humans. Perhaps the relevant CLSI Supplements targeting the human isolates are more appropriate to interpret the results. Please comment. 

Answer: Thank you for this question. We consider that if available, animal breakpoints should be used to interpret AST results for isolates of animal origin.

Comment: P12 L439-441. I am wondering which was the purpose of the calculation of MIC50 and MIC90. Usually, they are calculated in large surveys for the establishment of breakpoints. In similar papers, the characterization of the isolates as resistant or not is usually made. Please comment.  P12 L439-441.

Answer: Our aim was not to assignment breakpoints. However, the MICs 50 and 90 values were provided because it may be useful for other researchers, especially since there is still a problem with the availability of breakpoints for some animal pathogens. These data are more relevant to comparative studies than interpreting resistant/susceptible results alone.

Comment: P14 L540. Since there was this methodological flaw, I would not consider this appropriate.  

Answer: This sentence has been revised. (Page 15, Line 547-549)

Reviewer 2 Report

Very nice summary of the bacteria found at slaughter in LN.  Well done!   

The only significant comment that I have is on line 42 you list Streptococcus aureus. I think you mean Staph?? 

The paper is acceptable as written and the authors english is much better than my polish and is much better than I would do in a non-native language. But... it could benefit from a native english speaker.  A quick look over by a copy editor could make this paper much more readable. Not to change content but to improve the flow of the prose. 

Author Response

Comment: The only significant comment that I have is on line 42 you list Streptococcus aureus. I think you mean Staph?? 

Answer: Thank you. We have corrected this mistake.  (Page 1, Line 43)

Reviewer 3 Report

Figure 2, texts and numbers in the dendrogram are not precise. Please edit this picture.

Author Response

Comment: Figure 2, texts and numbers in the dendrogram are not precise. Please edit this picture.

Answer: Thank you. We have corrected this figure.

Reviewer 4 Report

The aim of this study was to evaluate the genetic diversity and antimicrobial resistance of Streptococcus spp. (n=48), S. aureus (n=5) and R. equi (n=17) strains isolated from swine lymph nodes with and without lesions. All isolates of S. dysgalactiae, S. aureus and R. equi were subjected to PFGE analysis. The number of strains analyzed in this study is relatively small, and the results obtained from this study are all known reports, lacking innovation. I don't think it can be published in this journal.

Minor editing of English language required.

Author Response

Comment: The aim of this study was to evaluate the genetic diversity and antimicrobial resistance of Streptococcus spp. (n=48), S. aureus (n=5) and R. equi (n=17) strains isolated from swine lymph nodes with and without lesions. All isolates of S. dysgalactiae, S. aureus and R. equi were subjected to PFGE analysis. The number of strains analyzed in this study is relatively small, and the results obtained from this study are all known reports, lacking innovation. I don't think it can be published in this journal.

Answer:  Thank you for your comment. The small number of isolates studied in this study results from the relatively low prevalence of the tested bacterial species. To our knowledge, this study reports for the first time the occurrence of these pathogens among swine lymph nodes in Poland. The manuscript has been corrected according to the comments of all reviewers.

Round 2

Reviewer 1 Report

I would like to thank the authors for their effort.

Reviewer 4 Report

Accept in present form.